# Enterprise Text-to-SQL: Are Agents Ready to Scale?

Tarık Buğra Akbaş[†]    Kemal Tarık Nehrozoğlu[†]    Pinar Karagoz[†]    Nesime Tatbul[*]

[†]METU, Türkiye    [*]Intel and MIT, USA

{tarik.akbas,tarik.nehrozoglu,pinark}@metu.edu.tr,tatbul@csail.mit.edu

## ABSTRACT

The integration of agentic architectures has significantly advanced the state-of-the-art in text-to-SQL tasks, enabling complex reasoning and self-correction through multi-step, modular workflows. However, the performance gains achieved by these systems, such as those utilizing ReAct-style loops, hierarchical delegation, and multi-agent collaboration, often come at a substantial computational and monetary cost that remains insufficiently quantified. This paper presents a comprehensive cost analysis of leading agentic text-to-SQL frameworks, evaluating the trade-offs between execution accuracy and operational expenditure. By profiling token consumption, GPU energy, and end-to-end SQL generation latency across the Spider, BIRD, and BEAVER benchmarks, we provide a comparative landscape of agentic efficiency.

## VLDB Workshop Reference Format:

Tarık Buğra Akbaş[†]    Kemal Tarık Nehrozoğlu[†]    Pinar Karagoz[†] Nesime Tatbul[*] . Enterprise Text-to-SQL: Are Agents Ready to Scale?. VLDB 2026 Workshop: NOVAS Workshop.

## VLDB Workshop Artifact Availability:

The source code, data, and/or other artifacts have been made available at https://github.com/TarikBugraAkbas/text2sql.

## 1 INTRODUCTION

Text-to-SQL, the task of translating a natural language question into an executable SQL query, has long been considered an important benchmark of practical natural language understanding. Its industrial relevance is clear: enabling non-expert users to query databases directly would unlock data access across enterprises, scientific institutions, and public services alike. With the emergence of large language models (LLMs), the field has experienced remarkable progress, with state-of-the-art systems now achieving execution accuracy above 90% on established benchmarks such as Spider [17].

This progress, however, has not come evenly. The leaderboard gains of recent years have been driven by increasingly elaborate pipelines: multi-step schema linking, query decomposition, self-correction loops, and multi-agent collaboration [13, 15]. Each additional stage improves accuracy but also multiplies the number of LLM calls, the tokens consumed, the latency incurred, and the energy drawn from the underlying hardware. The field has converged on accuracy as the dominant yardstick while treating cost as an afterthought.

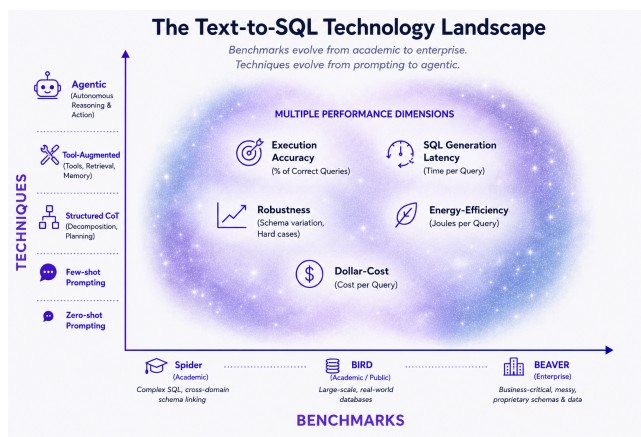

**Figure 1: Overview of text-to-SQL technology landscape.**

Figure 1 illustrates the evolving landscape of text-to-SQL technology. As benchmarks transition from academic environments to enterprise-scale complexity, methodologies are increasingly shifting toward agentic architectures. Consequently, the primary evaluation metrics are expanding beyond mere accuracy to encompass a diverse range of cost and efficiency factors. For example, recent literature has begun evaluating reasoning latency alongside token boundaries across specialized domain schemas [2, 3, 18]. However, a systematic cross-benchmark analysis that profiles evolving architectures directly at the hardware-level energy boundary remains absent.

This paper aims to address this gap, by evaluating four representative text-to-SQL methods of increasing complexity: a one-shot baseline (*FewShot*), a similarity-driven few-shot selector (*DAIL-SQL* [4]), a four-stage decomposition pipeline (*DIN-SQL* [13]), and a three-agent collaborative framework (*MAC-SQL* [15]); across three benchmarks of distinct character: Spider [17] and BIRD [10] academic benchmarks, and the BEAVER [1] enterprise benchmark built based on private production data warehouses. Crucially, we evaluate cost dynamics from an infrastructure perspective. Rather than relying solely on estimated API pricing, we instrument queries using hardware-level analytics to record the actual GPU energy consumed per inference pass, mapping energy expenditure alongside token counts and latency metrics to build a comprehensive cost profile for each pipeline paradigm.

The main contributions of this work can be listed as follows:

- **Comprehensive evaluation methodology.** To provide a holistic view of the state-of-the-art in text-to-SQL performance, we present an evaluation methodology that can objectively assess a wide spectrum of techniques (few-shot prompting, decomposition pipelines, and multi-agent frameworks) across diverse

benchmark complexities. Furthermore, we look beyond accuracy alone, mapping the tradeoffs between model performance and operational costs across three key dimensions: energy footprint, latency, and token consumption.

- **Hardware-level energy profiling.** We instrument every query individually with NVIDIA DCGM, reporting per-query GPU energy in joules. To our knowledge, this is the first hardware-level per-query GPU energy measurement study in text-to-SQL.
- **Accuracy/cost tradeoff analysis for open-source and proprietary models.** Our evaluation methodology enables a detailed comparative study of text-to-SQL methods and benchmarks, revealing critical tradeoffs that span across both accuracy and cost dimensions. We show how to tailor this analysis when using proprietary models, where additional metrics such as dollar-cost and cache-utilization also become relevant.
- **Extensible and reusable evaluation framework.** The proposed evaluation methodology is designed as an extensible framework, allowing for seamless integration of new metrics as the text-to-SQL landscape evolves. To support reproducibility and future research, we made our implementation publicly available and easily adaptable to other text-to-SQL studies.

## 2 RELATED WORK

The text-to-SQL landscape has been in rise for both research and industry, with contemporary solutions increasingly centered on Large Language Models (LLMs) [8, 9, 13, 14, 16]. While the literature is extensive [5, 11, 12], evaluation benchmarks remain heavily concentrated on correctness metrics such as Exact Match (EM) and Execution Accuracy (EX) [17, 19]. Systematic analysis of the computational and economic overhead of query generation remains a secondary focus, though recent studies have begun addressing this imbalance. For example, Zhang et al. incorporated monetary cost reporting for closed-source models in a finance-specific context [18], while Li et al. introduced a framework measuring GPU memory and latency alongside standard accuracy for PLM-based methods [7]. Furthermore, integrated metrics have emerged to reconcile performance with efficiency. The BIRD dataset introduced the Valid Efficiency Score (VES) to account for execution time [10], a metric later refined by Germán et al. into VES* and VCES to capture reasoning latency and end-to-end costs in ReAct-style architectures [3]. Expanding this scope further, the BIRD-INTERACT framework simulates multi-turn CRUD operations, evaluating success rates alongside rewards and monetary expenditure [6].

The body of research represented by DIN-SQL [13], DTS-SQL [14], and MAC-SQL [15] illustrates a paradigm shift in Text-to-SQL from monolithic generation to modular, agentic workflows that prioritize both reasoning depth and operational transparency. While DIN-SQL and DTS-SQL established the efficacy of multi-step decomposition for complex schemas, MAC-SQL expanded this to parallel multi-agent collaboration with interactive error handling. In [13], the authors provide a detailed cost-benefit analysis, highlighting that DIN-SQL's multi-step prompting significantly increases token cost compared to *vanilla* few-shot prompting. In [14], the *cost per successful query* is explicitly quantified. In [15], the authors claim that MAC-SQL reaches a point of diminishing returns in accuracy versus token expenditure.

While a growing body of research attempts to capture operational overheads, a holistic evaluation that simultaneously addresses benchmark complexity, methodological variations, and the accuracy/cost tradeoff remains absent. We bridge this gap by providing a comprehensive, multi-dimensional analysis.

## 3 EVALUATION METHODOLOGY

### 3.1 Text-to-SQL Benchmarks

We use a representative set of text-to-SQL benchmarks in our evaluation, which span a spectrum of schema and query complexity. This includes a purely academic benchmark (Spider [17]), an academic benchmark based on real-world data (BIRD [10]), and a more recently released enterprise benchmark based on larger scale real-world data and query workloads (BEAVER [1]).

**Spider** [17] is a cross-domain text-to-SQL benchmark comprising 1,034 development questions across 20 SQLite databases. It is widely used as a standard benchmark and serves here as a representative of well-structured, academic-style schemas.

**BIRD** [10] contains 1,534 development questions across 11 real-world databases. Compared to Spider, BIRD features larger schemas, dirtier data, and domain-specific *evidence* annotations (free-text hints about value mappings and business rules). We evaluate on the full development set and stratify results by the three official difficulty levels: *simple*, *moderate*, and *challenging*.

**BEAVER** [1] is an enterprise-grade benchmark spanning three private, large-scale data warehouses: DW, an Oracle academic warehouse with 97 tables and 1,530 columns; NW, a research-laboratory MySQL cluster with 366 tables across five databases; and SP, a housing-management MySQL system with 349 tables across two databases. We evaluate on a balanced subset of 621 queries sampled from the full benchmark: 371 from DW (121 real user logs + 250 synthesized), and 125 each from NW and SP, providing roughly equal coverage across warehouses and query types. Our evaluation uses two BEAVER settings: *BEAVER S1* provides gold tables, column mappings, and join keys as oracle hints, isolating generation cost from retrieval difficulty; *BEAVER S0* is fully end-to-end: tables are retrieved with Qwen3-Embedding-8B (top-50) and re-ranked by Qwen3-Reranker-8B (top-15).

### 3.2 Agentic Text-to-SQL Methods

We implement and evaluate four Text-to-SQL generation strategies of increasing pipeline complexity. We use FewShot as a complexity baseline and have chosen the remaining three methods from top scoring standalone methods from BIRD and Spider benchmarks.

**FewShot** is a single-call baseline. The model receives the full database schema, one hand-crafted example question-SQL pair, and the target question (plus evidence for BIRD), and generates SQL in one pass. It establishes the cost lower bound and the accuracy floor.

**DAIL-SQL** [4] replaces the fixed example with dynamically selected few-shot demonstrations. For each query, we encode the question using `sentence-transformers/all-mpnet-base-v2` and retrieve the $k$=3 most similar questions from the training set by cosine similarity, excluding examples from the same database. On BEAVER, where no annotated Q-SQL training pairs exist, we use Qwen3-Embedding-8B over a hand-curated example pool instead. Like FewShot, only a single LLM call is made. When this paper

was written, DAIL-SQL was in 2nd place of the Spider leaderboard and 61st place in the BIRD Validation Efficeincy Score (VES) leaderboard.

**DIN-SQL** [13] is a four-stage pipeline: *schema linking*, *query classification*, *SQL generation* with chain-of-thought reasoning, and *self-correction*. Each stage is a separate LLM call, yielding a minimum of four calls per query. When this paper was written, DIN-SQL was in 4th place of the SPider leaderboard and 63rd place in BIRD VES leaderboard.

**MAC-SQL** [15] is a three-agent collaborative framework. A *Selector* agent identifies relevant tables and columns; a *Decomposer* agent generates SQL with chain-of-thought reasoning over the filtered schema; and a *Refiner* agent self-corrects on execution errors up to three times. MAC-SQL makes two to five LLM calls per query. When this paper was written, MAC-SQL was in 52nd place in the BIRD VES leaderboard.

## 3.3 Evaluation Metrics and Cost Measurements

We report SQL generation accuracy in *Execution Accuracy* (EX), where a generated SQL is counted correct if its result set matches the gold SQL result set after order-independent normalization.

We measure cost along three dimensions. *Token cost* is the sum of input and output tokens reported by the Ollama API for each LLM call, aggregated per query. *Latency* is the wall-clock time from first LLM call to final SQL. *GPU energy* is measured using NVIDIA DCGM sampled at query boundaries, yielding per-query consumption in Joules (J). This hardware-level measurement captures actual inference cost independent of pricing models, and is to our knowledge the first hardware-level per-query GPU energy measurement study in Text-to-SQL.

For the OpenAI API experiments, we additionally report *dollar cost per query* (billed input + output tokens at GPT-5-mini list prices) and *APC cache miss rate*: the fraction of input tokens *not* served from OpenAI's Automatic Prompt Cache, where a miss means the full prompt was re-evaluated on the server at full price.

## 4 EXPERIMENTAL ANALYSIS

### 4.1 Experiment Environment and Models

In our experiments, we primarily use qwen2.5-coder:32b, served locally via Ollama on a single NVIDIA RTX 6000 Ada (49 GB). Temperature is set to 0 throughout for deterministic outputs. Additionally, GPT-5-mini is used through the OpenAI API to measure dollar cost and prompt caching for our last set of experiments.

### 4.2 Execution Accuracy across Benchmarks

Figure 2 reports EX Accuracy % across all three benchmarks, including both BEAVER settings ($S_0$ and $S_1$, 621 queries each). A few findings stand out:

First, **method ranking is not stable across benchmarks**. DAIL-SQL leads on Spider (78.7%) while FewShot (76.0%) finishes second; MAC-SQL is the top method on both BIRD (53.5%) and BEAVER $S_1$ (12.7%) yet ranks last on Spider (74.0%). DAIL-SQL (50.7% on BIRD) tracks MAC-SQL closely, finishing just 2.8 pp behind at one-quarter of its token cost. DIN-SQL trails all methods on every benchmark.

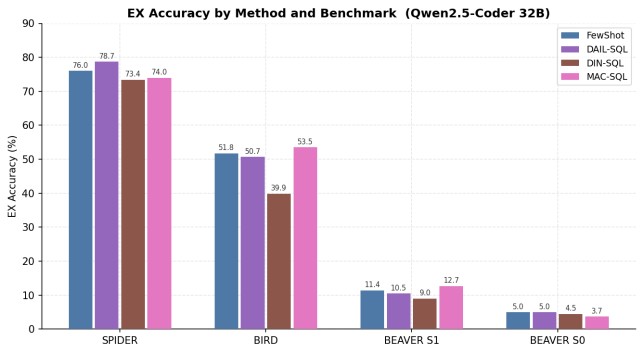

**Figure 2: Execution accuracy across benchmarks using Qwen2.5-Coder 32B**

Second, **BEAVER reveals an enterprise performance floor**. All four methods collapse into a narrow 9.0–12.7% EX band on the enterprise warehouses under $S_1$ regardless of pipeline complexity, and further degrade to a 3.7–5.0% band under the end-to-end $S_0$ setting. Enterprise schema complexity and retrieval constitute qualitative barriers that academic benchmarks do not expose.

> **Key Insight:** *No single method dominates across all three benchmarks. As benchmark complexity grows, agentic methods gain an accuracy edge (MAC-SQL leads on BIRD and BEAVER $S_1$); but on truly enterprise end-to-end workloads (BEAVER $S_0$), no method yet achieves meaningful accuracy.*

### 4.3 Cost Profile across Benchmarks

Figure 3 shows per-metric absolute cost (GPU energy, latency, tokens) across all benchmark/method combinations as a color-normalized heatmap.

The heatmap reveals a cost split on BEAVER $S_1$: in latency, multi-step methods pay a heavy overhead—MAC-SQL takes 59 s/q and DIN-SQL 86 s/q due to sequential agent calls (vs. FewShot's 17 s/q). In tokens, however, **both DAIL-SQL and MAC-SQL are cheaper than FewShot**: DAIL-SQL retrieves only a short hand-curated example (2.6K tok/q) and MAC-SQL's Selector prunes the schema (5.4K tok/q) vs. FewShot's full 97-table schema (7,286 tok/q average).

On BIRD and Spider the picture is reversed: every method above FewShot costs more in both latency and tokens, with DIN-SQL reaching 27 s/q and 3.9K tok/q on BIRD (vs. FewShot's 1.6 s/q and 0.6K tok/q).

Three patterns stand out in the heatmap. First, costs generally scale with benchmark difficulty, but DIN-SQL is the notable exception: its GPU energy and latency are lower on BIRD (7.1 kJ, 27 s) than on Spider (11 kJ, 44 s), suggesting that schema structure rather than query count drives its overhead. Second, DAIL-SQL is consistently more efficient than FewShot, most starkly on BEAVER where retrieved examples replace the full 97-table schema (2.6K vs. 7.3K tok/q under $S_1$). Third, MAC-SQL's costs are lower on BIRD than on Spider (3.4 kJ, 13 s vs. 4.7 kJ, 18 s) before jumping sharply on BEAVER ($S_0$: 32 kJ, 176 s/q).

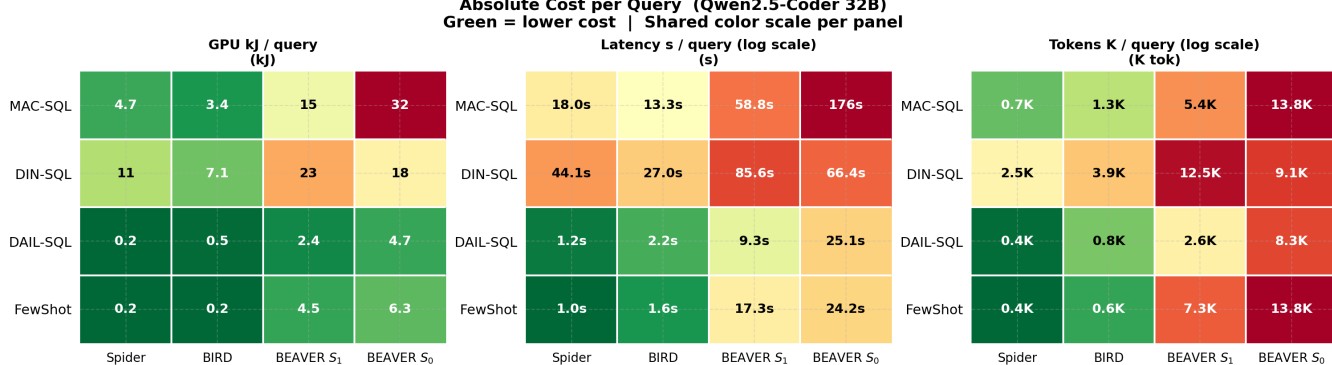

**Figure 3: Absolute cost heatmap across benchmarks and methods. Each panel shares a single color scale (log-normalized for latency and tokens). Green = lower cost.**

---

> **Key Insight:** *Costs scale with benchmark difficulty as a general rule, but schema exposure overrides this trend. Methods that limit schema input (DAIL-SQL via example retrieval, MAC-SQL via its Selector) consistently beat the trend, while DIN-SQL's anomaly (higher cost on Spider than BIRD) reveals that schema structure matters more than query count.*

### 4.4 Accuracy/Cost Tradeoff

Figure 4 breaks down cost by query outcome across all benchmarks. Each method has two independent bars: the solid bar is the average cost over correct queries and the hatched bar is the average cost over incorrect queries. A consistent pattern emerges: **incorrect queries always cost more than correct ones**, across every method and every cost metric. This is not explained by extra retries. DIN-SQL issues exactly four LLM calls per query regardless of outcome, yet the gap persists. The explanation is *selection bias*: the queries a model fails on tend to be structurally harder (larger schemas, more complex joins, longer prompts), and those same properties inflate GPU energy, latency, and token count independently of the method's control flow. MAC-SQL additionally issues extra refinement calls on failed queries, compounding the gap.

Figure 5 plots EX% against each cost dimension across all benchmarks. On BEAVER $S_1$, DAIL-SQL occupies the Pareto efficiency frontier: at 2.4 kJ/q, 9.3 s/q, and 2.6K tok/q, it is the only method both cheaper and faster than FewShot (4.5 kJ/q, 17.3 s/q, 7.3K tok/q). DIN-SQL is strictly dominated at 23 kJ/q, 86 s/q, and 12.5K tok/q. Cheaper methods are not necessarily less accurate on BEAVER. DAIL-SQL achieves $\sim$ 8% EX under $S_1$ while consuming only 2.4 kJ/q GPU energy (vs. FewShot's 4.5 kJ) and 2.6K tok/q (vs. 7.3K), sitting on the Pareto frontier across all cost dimensions. MAC-SQL pays 59–176 s/q in latency (vs. FewShot's 17–24 s) for marginal accuracy gains, yielding poor cost-per-correct-query efficiency. The correct-vs-incorrect cost gap (Figure 4) is an artifact of query difficulty distribution, not method behavior; a distinction that matters when interpreting per-query cost averages.

> **Key Insight:** *"Less agentic" methods (FewShot and DAIL-SQL) dominate the efficiency frontier across all benchmarks and cost dimensions, an advantage not visible when accuracy and cost are examined separately, only when considered jointly.*

### 4.5 Accuracy/Cost Tradeoff (Proprietary Model)

Running all four methods via the OpenAI API adds two observables absent from local runs: *dollar cost* and *Automatic Prompt Caching* (APC). APC serves the longest stable prefix of a prompt from a server-side cache at lower prices; it activates only for prompts exceeding 1,024 tokens, making it irrelevant for short academic prompts but significant on BEAVER where schema-heavy inputs routinely surpass this threshold. Methods with fixed prompt structure benefit automatically, while fully dynamic pipelines pay full price every call. Figure 6 summarizes results across all three benchmarks.

Accuracy on Spider and BIRD remains competitive with local Qwen2.5-Coder 32B. DAIL-SQL leads on Spider (74.6%), beating FewShot (73.5%) at equal cost and half the latency, and leads on BIRD (50.3%) while still outrunning DIN-SQL (41.5%) despite a simpler, single-call design. On BEAVER, EX collapses to 3.5–10.3% across all methods (MAC-SQL best at 10.3%), reaffirming that enterprise schema complexity is the binding constraint regardless of provider or pipeline design.

The Cache Miss Rate column reveals how prompt architecture interacts with caching. FewShot achieves a 26% miss rate on BEAVER $S_1$ because its instruction block and fixed example are identical across all 621 queries; only the per-query schema tail escapes the cache. DIN-SQL's miss rate is 59% via a different mechanism: its four sequential calls all carry the same schema from call 1, so calls 2–4 hit the cache for that repeated block (*intra-query* reuse, not cross-query). DAIL-SQL hits 100% miss rate on Spider and BIRD because similarity-selected examples change with every query, collapsing the stable prefix; on BEAVER its fixed curated example reduces this to 87–90%. MAC-SQL exceeds 99% miss rate under $S_1$ as the Selector prunes a unique column subset per query and the Refiner embeds execution error messages, making every call effectively novel; under $S_0$ it falls to 88%. Cache miss rates reach 100% for all methods on Spider, where 20 different databases force the schema (the dominant token block) to change every query.

The primary cost driver across benchmarks is **schema volume**, not method complexity. FewShot costs 32× more per query on BEAVER $S_0$ than on Spider ($5.17×10^{-3}$ vs $0.16×10^{-3}$) purely from injecting 15 retrieved tables. Switching to gold hints ($S_1$) cuts cost

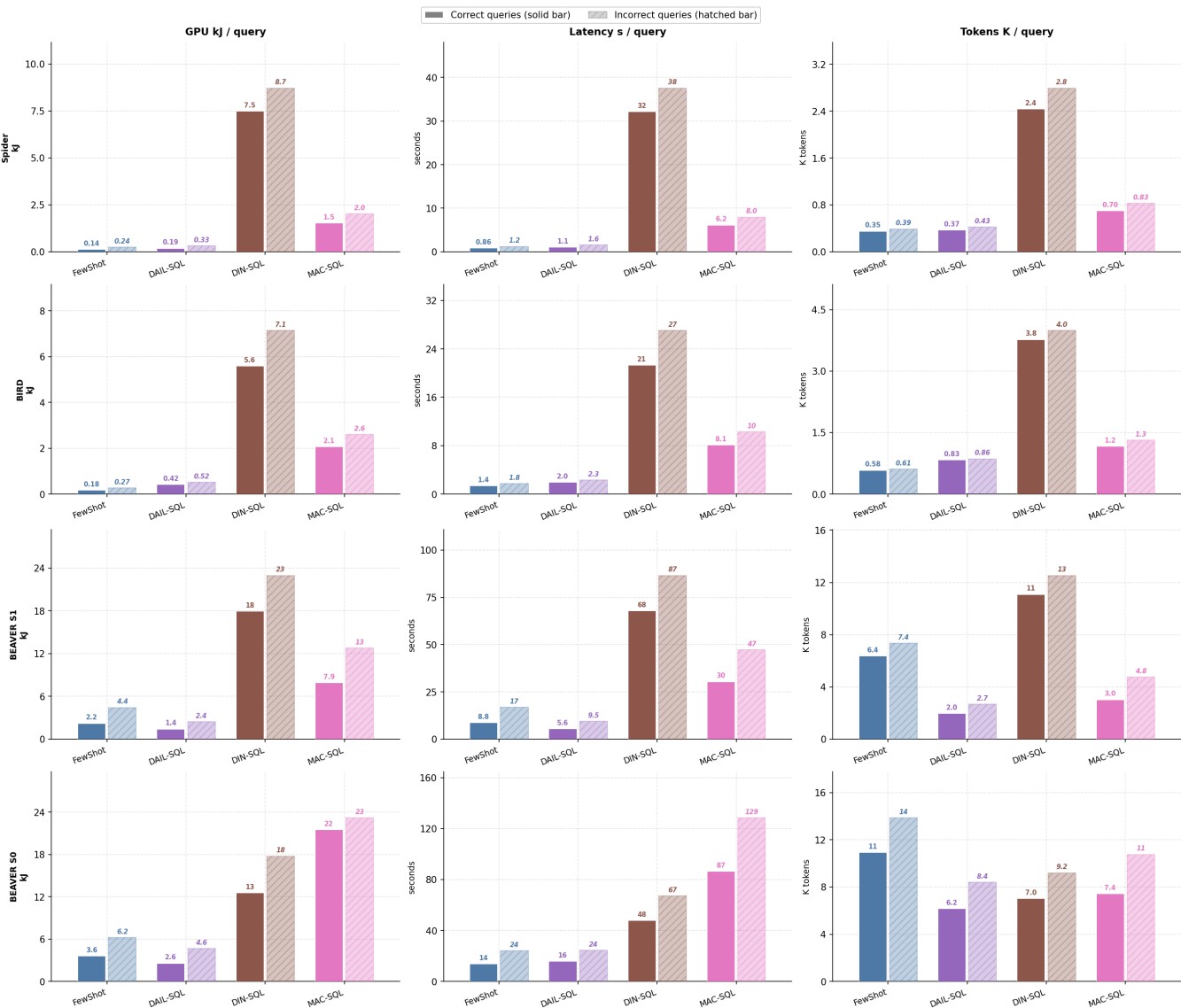

**Figure 4: Average cost per query broken down by outcome (Qwen2.5-Coder-32B). Rows = benchmarks; columns = GPU energy, latency, tokens. For each method, the left (solid) bar shows the average cost for correct queries and the right (hatched) bar shows the average cost for incorrect queries; the two bars are independent and share the same y-scale.**

4.5× for FewShot and 1.75× for MAC-SQL, making table retrieval the dominant expenditure in end-to-end settings.

**Key Insight:** *Our findings generalize across model providers: DAIL-SQL's efficiency-edge and BEAVER's accuracy-collapse both replicate on GPT-5-mini, confirming the bottleneck is schema complexity, not model choice. The one proprietary-only lever is prompt caching, which discounts cost for fixed-prefix methods like FewShot, but this alone is not sufficient to recover the loss in accuracy.*

## 5 CONCLUSION

This study introduces a methodology for analyzing accuracy-cost tradeoffs within the rapidly evolving text-to-SQL landscape, specifically tracking the shift toward agentic architectures and enterprise-scale benchmarks characterized by complex schemas and queries. Our empirical evaluation spans three benchmark datasets (ranging from academic to enterprise-scale) and four distinct methodologies (ranging from single LLM calls to multi-call cooperative agentic architectures). The resulting data quantifies the operational overhead inherent to each approach and isolates the underlying cost drivers,

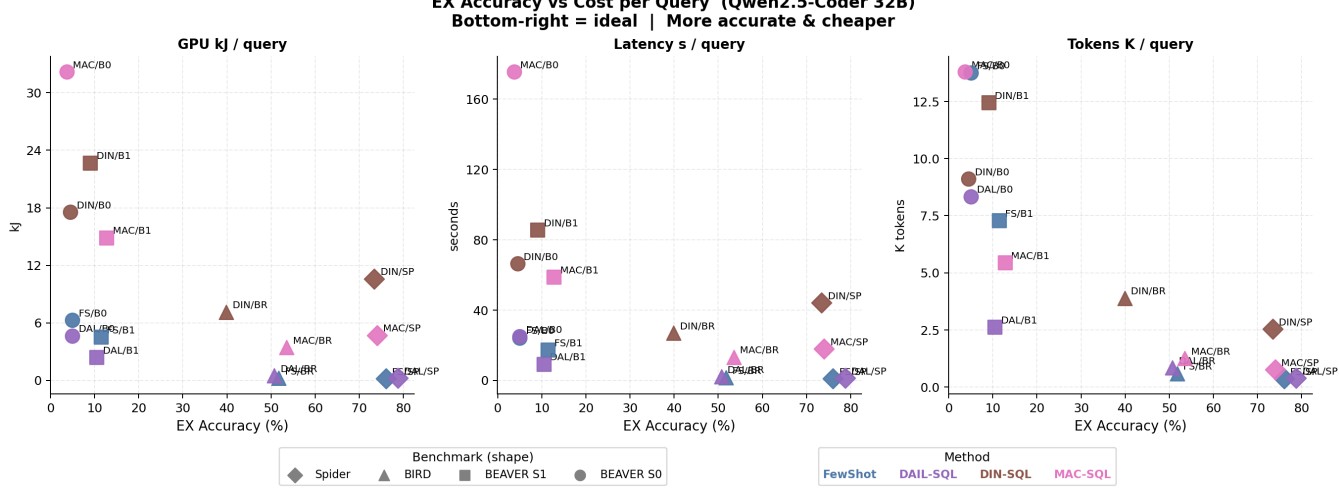

Figure 5: Execution accuracy (EX%) vs. cost per metric across benchmarks. Each panel shows one cost dimension (GPU kJ/query, latency, tokens/query). DAIL-SQL dominates the efficiency frontier on BEAVER; FewShot dominates on Spider.

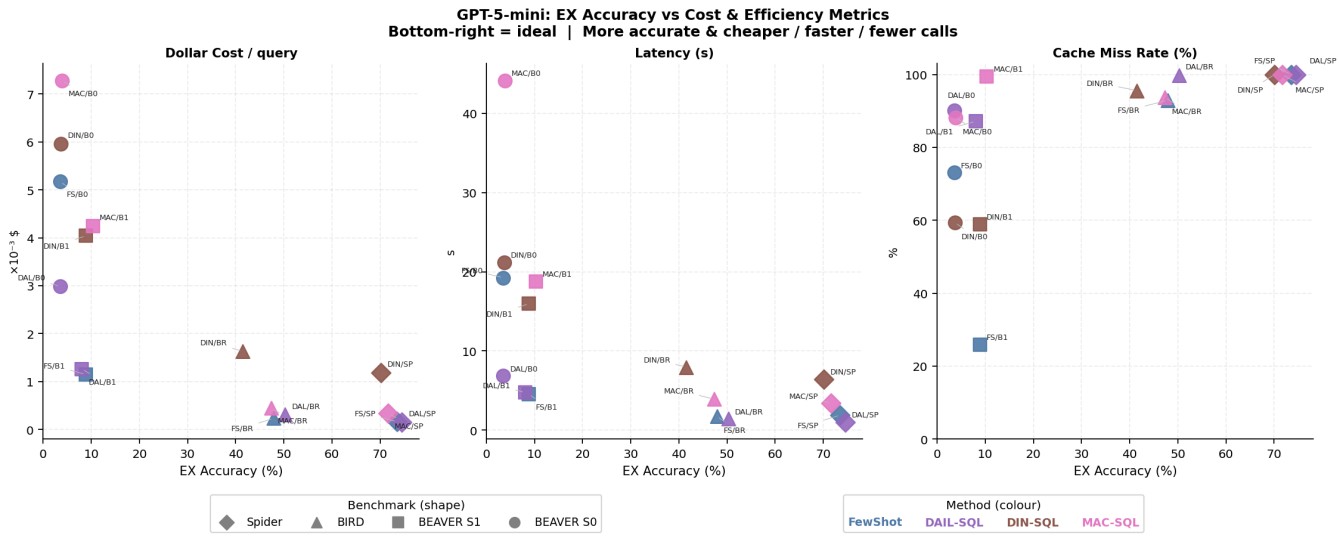

Figure 6: GPT-5-mini across benchmarks (Spider 1,034 q; BIRD 1,534 q; BEAVER 621 q). Each point is one method/benchmark pair. Shape encodes benchmark; color encodes method.

offering actionable insights for designing future enterprise-grade text-to-SQL solutions. A promising avenue for future work is the integration of our evaluation methodology into standard text-to-SQL benchmarking systems, establishing a more multi-dimensional evaluation standard for the research community.

## ACKNOWLEDGMENTS

We thank Peter Baile Chen and Michael Stonebraker for their feedback and support. This research received the support of the EXA4MIND project, funded by the European Union's Horizon Europe Research and Innovation Programme, under Grant Agreement No 101092944. Views and opinions expressed are however those of the author(s) only and do not necessarily reflect those of the European Union or the European Commission. Neither the European Union nor the granting authority can be held responsible for them. This work is also partially funded by METU under the grant no. ADEP-312-2024-11484. LLMs were utilized to generate Figure 1, and to improve the text and the code.

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
