# OpenReview forum: "Enterprise Text-to-SQL: Are Agents Ready to Scale?"
_VLDB.org/2026/Workshop/NOVAS — NOVAS 2026_

### Official Review · Reviewer_12sK · 2026-06-30

**Confidence:** 4

**Improvement Opportunities:**

I1) Although FewShot ranks second on each dataset in Figure 2, it is costly because it includes the full database schema, which can be large in datasets such as BIRD or BEAVER. Is this really a lower bound for cost estimation and accuracy? A hybrid approach (FewShot + schema pruning) using top-k retrieval over table and attribute names could reduce the context size and cost, while still being effective.

I2) Figure 3 shows a relationship between GPU cost and latency: both seem to capture a similar trend, namely that higher execution time corresponds to higher GPU consumption. This relationship does not appear to be directly driven by token consumption, but rather by the complexity of the database schema, the query, and the NL2SQL method. This raises the question of whether token-based API pricing accurately reflects the underlying computational cost, especially when methods with similar token counts exhibit very different latency and energy profiles as shown, for example, by FewShot on BEAVER S0.  It would be interesting to report an heatmap similar to Figure 3 for the experiments in Section 4.5, including token count, billed cost, latency, and cache miss rate. Although API latency is only an imperfect proxy for GPU usage, such an analysis could clarify how monetary cost relates to actual computational effort, and maybe estabilish a research direction as reported in S1.

**Minor Comments:**

M1) The citation style could be improved. Ideally, a sentence containing a citation should remain readable even without reading the citation itself, for example on page 2, at the end of the left column.

**Short Summary:**

The paper studies the Text-to-SQL problem by evaluating different Text-to-SQL approaches. While most existing works focus mainly on execution accuracy, this paper also considers additional dimensions such as cost, latency, energy consumption, database complexity, and query complexity. Reasoning across all these dimensions can help enterprises better understand the costs associated with specific gains in accuracy.

**Strong Points:**

S1) The experimental methodology is very interesting, since it enables a multidimensional evaluation and provides useful insights for the practical deployment of Text-to-SQL methods in enterprise environments. It also enables further studies on how to choose between pay-as-you-go strategies based on token consumption and time consumption.

S2) The accuracy/cost trade-off analysis shows that agentic methods are not always the best choice under these criteria. This could open up a future research direction aimed at inferring, based on the schema and a sample of real query workloads, which NL2SQL strategy should be applied.

S3) The paper is well-written and easy to follow.

---

### Official Review · Reviewer_Hqcs · 2026-07-10

**Confidence:** 4

**Improvement Opportunities:**

1.The BEAVER results deserve more diagnostic analysis. The paper shows that all methods perform poorly on BEAVER, especially in S0. This is one of the most important findings, but the paper could better explain why. For example, are failures mostly caused by table retrieval, column linking, join path discovery, SQL syntax, missing business logic, or value grounding? A small error taxonomy would make the enterprise-performance-floor claim much more actionable.
2.The comparison between methods may be affected by implementation choices. FewShot, DAIL-SQL, DIN-SQL, and MAC-SQL differ not only in agentic complexity but also in prompt structure, schema exposure, example selection, and retry behavior. The paper should clarify how faithfully each method was implemented and whether prompts were tuned equally.
3.The energy measurement methodology needs more detail. The paper states that GPU energy is measured with NVIDIA DCGM at query boundaries. It would be useful to specify sampling frequency, whether idle baseline energy is subtracted, how overlapping processes are controlled, and whether database execution energy is included or only LLM inference energy.

**Minor Comments:**

The discussion of proprietary models could be expanded cautiously. GPT-5-mini is useful as an API-based comparison, but one proprietary model may not fully represent closed-source deployment.

**Short Summary:**

This paper studies the accuracy-cost tradeoff of agentic text-to-SQL systems, especially in enterprise-scale settings. The authors evaluate four representative methods on three benchmarks, measuring their execution accuracy, token consumption, SQL generation latency, and hardware-level GPU energy using NVIDIA DCGM. The central finding is that more agentic workflows do not uniformly dominate. While MAC-SQL performs best on BIRD and BEAVER S1 in terms of accuracy, simpler methods such as FewShot and DAIL-SQL often lie on the efficiency frontier. The paper also argues that enterprise schema complexity and schema retrieval, rather than model choice alone, are major bottlenecks for scalable text-to-SQL.

**Strong Points:**

1.The paper addresses an important and timely problem. Many text-to-SQL papers focus mainly on execution accuracy, while real enterprise deployment also depends on latency, token cost, and infrastructure cost.
2.The evaluation is broader than a standard accuracy-only comparison. This helps show that rankings on academic benchmarks do not necessarily transfer to enterprise workloads. The inclusion of BEAVER is particularly valuable because it reveals that enterprise schema complexity can collapse performance even when methods perform well on Spider or BIRD.
3.Hardware-level GPU energy measurement is a useful contribution. Measuring per-query GPU energy with NVIDIA DCGM is a concrete and practical addition to text-to-SQL evaluation.

---

### Official Review · Reviewer_ziSc · 2026-07-10

**Confidence:** 3

**Improvement Opportunities:**

O1: The paper is primarily an experimental evaluation paper and lacks a strong technical contribution. While the empirical study is valuable, the paper does not propose a new method, technique, or evaluation metric.

O2: Figure 1 should be significantly improved. It serves as a central figure and is positioned on the first page, yet it appears clearly AI-generated and does not meet the quality expected for a key explanatory figure. The resolution should be improved, and the figure should be simplified and reorganized, as it currently feels overcrowded and difficult to interpret.

O3: Figure 2 should be made colorblind-friendly.

O4: The discussion of evaluation metrics is insufficient. The paper correctly notes limitations of traditional exact-match style evaluation but relies on "Execution Accuracy" as the primary accuracy metric. Execution Accuracy limitations deserve more discussion. Additional accuracy measures would strengthen this portion of paper. In particular, I expected some form of fuzzy or semantic similarity metric alongside Execution Accuracy. Execution Accuracy can be highly biased. For example, two semantically different queries that both return an empty result set would be considered equivalent under this metric. In contrast, a generated query containing only a minor typo that could easily be corrected by an end user might receive a score of 0 if the query fails to execute. These trade-offs should be clearly discussed, and alternative metrics should at least be considered.

**Minor Comments:**

n/a

**Short Summary:**

This paper presents an experimental study of modern text-to-SQL systems, focusing on the trade-offs between accuracy and operational cost. The authors evaluate several representative approaches, ranging from few-shot prompting to agentic multi-step frameworks, across academic and enterprise benchmarks.

**Strong Points:**

S1: Agentic text-to-SQL are highly relevent to the NOVAs community and VLDB at large.

S2: The evaluation is comprehensive, covering multiple benchmarks and methods of varying complexity.

S3: The paper is well structured, readable and provides fairly interesting key insights.

---

### Decision · Program_Chairs · 2026-07-16

**Decision:**

Accept

**Comment:**

This paper provides a comprehensive evaluation of the accuracy, latency, token cost, and energy consumption of modern Text-to-SQL systems across academic and enterprise benchmarks. The multidimensional analysis offers useful deployment insights, particularly by showing that more agentic methods do not always provide the best cost-quality trade-off. The findings are highly relevant to NOVAS, and we hope they motivate further discussion on efficient enterprise Text-to-SQL systems.